# Low-Dose High-Resolution Photon-Counting CT of the Lung: Radiation Dose and Image Quality in the Clinical Routine

**DOI:** 10.3390/diagnostics12061441

**Published:** 2022-06-11

**Authors:** Matthias Michael Woeltjen, Julius Henning Niehoff, Arwed Elias Michael, Sebastian Horstmeier, Christoph Moenninghoff, Jan Borggrefe, Jan Robert Kroeger

**Affiliations:** Department of Radiology, Neuroradiology and Nuclear Medicine, Johannes Wesling University Hospital, Ruhr University Bochum, 44801 Bochum, Germany; julius.niehoff@muehlenkreiskliniken.de (J.H.N.); arwed.michael@muehlenkreiskliniken.de (A.E.M.); sebastian.horstmeier@muehlenkreiskliniken.de (S.H.); christoph.moenninghoff@muehlenkreiskliniken.de (C.M.); jan.borggrefe@muehlenkreiskliniken.de (J.B.); janrobert.kroeger@muehlenkreiskliniken.de (J.R.K.)

**Keywords:** computed tomography, CT, photon counting detector, low dose chest CT, low dose lung CT, LD-HR CT, PCCT vs. EID

## Abstract

This study aims to investigate the qualitative and quantitative image quality of low-dose high-resolution (LD-HR) lung CT scans acquired with the first clinical approved photon counting CT (PCCT) scanner. Furthermore, the radiation dose used by the PCCT is compared to a conventional CT scanner with an energy-integrating detector system (EID-CT). Twenty-nine patients who underwent a LD-HR chest CT scan with dual-source PCCT and had previously undergone a LD-HR chest CT with a standard EID-CT scanner were retrospectively included in this study. Images of the whole lung as well as enlarged image sections displaying a specific finding (lesion) were evaluated in terms of overall image quality, image sharpness and image noise by three senior radiologists using a 5-point Likert scale. The PCCT images were reconstructed with and without a quantum iterative reconstruction algorithm (PCCT QIR+/−). Noise and signal-to-noise (SNR) were measured and the effective radiation dose was calculated. Overall, image quality and image sharpness were rated best in PCCT (QIR+) images. A significant difference was seen particularly in image sections of PCCT (QIR+) images compared to EID-CT images (*p* < 0.005). Image noise of PCCT (QIR+) images was significantly lower compared to EID-CT images in image sections (*p* = 0.005). In contrast, noise was lowest on EID-CT images (*p* < 0.001). The PCCT used significantly less radiation dose compared to the EID-CT (*p* < 0.001). In conclusion, LD-HR PCCT scans of the lung provide better image quality while using significantly less radiation dose compared to EID-CT scans.

## 1. Introduction

Computed tomography (CT) of the chest is the most common imaging technique for analyzing abnormalities of the lung parenchyma associated with numerous diseases. Among others, it offers the possibility to assess lung lesions in terms of location, size and density. However, with the increasing quantity of examinations over the past decades, CT examinations are contributing significantly to the overall radiation dose exposure in the population [1].

The introduction of low-dose CT protocols with reduced radiation dose exposure for the examination of the chest in the 1990s made it possible to achieve good image quality without significantly sacrificing image sharpness and visualization of different lesions as well as parenchymal structures [2]. In the past decades, multiple studies underlined the value of low-dose CT scan protocols by protecting patients from high radiation dose exposure while maintaining good image quality [3,4,5]. This applies to CT examinations of various parts of the human body [6,7,8]. However, in the setting of interstitial lung disease, the evaluation of very small pulmonary changes (e.g., ground glass opacities, reticulation) requires a high spatial resolution. Thus, low-dose examinations with a high resolution are needed.

Today, several approaches exist that can lead to a significantly reduced radiation dose—particularly in low-dose CT examinations. Among others, these approaches include automatic tube current modulation and the use of lower tube potential. Furthermore, contemporary techniques, such as iterative reconstructions (IR), are applied routinely. IR methods create a simulated image and repeatedly compare this image with the measured projection data for correction of the simulated image [9]. With technological advancements—especially in computational power—different iterative reconstruction techniques are being used more frequently in the clinical routine, displacing filtered back projection. The majority of studies show a significant decrease in radiation dose while maintaining subjective and objective image quality when using IR methods [10].

A milestone in CT technology was the development of spectral CT scanners, accomplished by different technological approaches, such as dual-source or dual-layer CT [11]. Although the first studies reported on an increase in radiation dose compared to single-energy CTs [12], various hard- and software updates as well as advances in iterative reconstruction techniques have ultimately led to a decrease in overall radiation dose [13].

In 2021, the introduction of the first clinical photon counting CT (PCCT, Naeotom Alpha, Siemens Healthineers, Erlangen, Germany) with quantum technology promised to be the next major step in CT technology by further increasing spectral imaging capabilities. Conventional, energy-integrating detectors (EIDs) absorb X-ray photons in a scintillator material and convert these photons into visible light, which ultimately is measured by a photodiode generating a proportional electric signal. In contrast, photon counting detectors (PCDs) consist of a single semiconductor layer that converts individual X-ray photons directly into an electric signal [14,15]. In addition to their spectral imaging capacities, PCDs offer a higher spatial resolution and a reduction in radiation exposure. This results from a minimization of electronic noise offering further improvement for low-dose CT scans of the lung [14,16]. Along with the novel PCCT, new iterative reconstruction algorithms—known as quantum iterative reconstruction (QIR)—were introduced. These QIR algorithms were developed to meet the increased hard- and software requirements of the PCCT (e.g., in terms of data complexity, spectral information and noise profile) [17]. At the current state, four levels of QIR are available. Sartoretti et al. provide further information on these novel QIR algorithms [18]. Both preclinical studies using PCCT prototypes and early studies with the first PCCT scanner that is approved for clinical use show promising results [15,16,19,20,21].

The purpose of the present study is to compare the performance of the first PCCT approved for clinical use and a standard CT scanner using an EID in terms of radiation dose parameters as well as qualitative and quantitative image quality. The subject of this study is LD-HR CT scans of the chest.

## 2. Materials and Methods

### 2.1. Patient Population

The study was conducted according to the guidelines of the Declaration of Helsinki and approved by the institutional review board. Informed consent was waived due to the retrospective study design. All scans were performed for diagnostic use with clinical standard protocols. Patient data were anonymized.

In total, 29 patients (15 male and 14 female) who underwent an LD-HR chest CT with the PCCT between September 2021 and January 2022 in the clinical routine were retrospectively included in this study. The patients were chosen as they had undergone an LD-HR chest CT with a standard EID-CT scanner (S64, Somatom Sensation 64, Siemens Healthineers, Erlangen, Germany) in the past. These previous CT examinations were used for comparison. Only patients that did not show a significant change in pulmonary findings (as per the clinical radiological report) were included. Patients were not preselected regarding age, sex, weight or other characteristics.

### 2.2. CT Protocols and Image Acquisition

All patients underwent non-enhanced LD-HR CT scans of the chest, with either the PCCT (software version Syngo CT VA40, Siemens Healthineers, Erlangen, Germany) or the S64.

Examinations were performed in supine and in prone position. The scan parameters of the PCCT were as follows: tube voltage = 120 kVp, detector configuration = 144 mm × 0.4 mm, automatic tube current modulation, image quality level (IQ-level) = 70, pitch = 1.2 and gantry rotation time = 0.25 s. The scan parameters of the S64 were as follows: tube voltage = 120 kVp, detector configuration = 32 mm × 0.6 mm, no automatic tube current modulation, reference Q = 50 mAs, pitch = 1.4 and gantry rotation time = 0.5 s.

The reconstructed images were analyzed retrospectively using a standard PACS system (IMPAX 6.7, Agfa Healthcare, Mortsel, Belgium). All images were reconstructed in axial view with the same slice thickness (1 mm) on both CT scanners. The PCCT images were reconstructed with quantum iterative reconstruction (QIR, level Q2) and without QIR. The S64 images were reconstructed without iterative reconstruction algorithms. Consequently, 6 images of each patient were compared at the same anatomical position: 3 images pictured the whole lung (Figure 1a–c), while the other 3 images showed an enlarged image section displaying a specific finding (lesion) (Figure 1d–f). The images of the whole lung were leveled at the carina while the leveling of image sections varied between patients in accordance to the anatomic position of the displayed lung lesion. Window settings were constant in all images (center: −500 and width: 1700), which represents the standard window settings used at our institution.

### 2.3. Quantitative Image Analysis

Regions of interest (ROI) were placed in the lung parenchyma at the exact same position of all 3 images. The size of the ROIs was as large as possible, ensuring that large or dense structures, such as bronchi and lesions, were not included. The signal of an ROI was defined as the mean density in Hounsfield units (HU). Image noise was defined as the standard deviation (SD) of ROI density measurements. The signal-to-noise ratio (SNR) was calculated by dividing the mean density in HU in each ROI by its SD.

### 2.4. Radiation Dose

The computed tomography dose index (CTDI) and the dose length product (DLP) of each CT scan were collected. The effective dose (ED) was calculated with an application that allows the calculation of all dose quantities of practical value for patients undergoing CT examinations as described in detail by Stamm et al. [22].

The body weight of each patient required for the calculation of the effective radiation dose was obtained for PCCT examinations and accepted for the calculation of radiation doses of EID-CT examinations.

### 2.5. Qualitative Image Analysis

In total, 174 images (29 patients with 6 images each) were analyzed by 3 senior radiologists in terms of general image quality (1—deficient, 2—sufficient, 3—satisfactory, 4—good, 5—very good), image noise (1—distinct, 2—increased, 3—moderate, 4—little, 5—hardly) and image sharpness (1—deficient, 2—sufficient, 3—satisfactory, 4—good, 5—very good) using a 5-point Likert scale. The readers assessed the image quality parameters subjectively. The basis for the evaluation was the clinical assessability. The CT scans were anonymized, blinded and presented to the reader in a random order to impede conclusions on the used CT scanner and to prevent a comparison of images within a patient case.

### 2.6. Statistical Analysis

Established software packages were used for the statistical analysis (SPSS Statistics 28, IBM, Armonk, NY, USA; Excel 2016, Microsoft, Redmond, WA, USA; R Core Team (2021). R: A language and environment for statistical computing. R Foundation for Statistical Computing, Vienna, Austria. URL https://www.R-project.org/ (accessed on 26 October 2021); RStudio Version 1.4.1106). All available data are presented as mean ± SD if not stated otherwise.

Differences between EID, PCCT (QIR−) and PCCT (QIR+) in terms of qualitative image analysis were tested with the Wilcoxon test for connected samples. Differences in noise and SNR as well as dose between EID and PCCT scans were calculated using a two-sided paired *t*-test. Statistical significance was assumed for *p*-values < 0.05.

## 3. Results

### 3.1. Patient Population

The mean age of all patients was 67.7 years (range: 33–82 years); the mean age of male patients was 66.2 years (range: 33–82 years) and the mean age of female patients was 69.2 years (range: 56–78 years). The mean time span between both CT scans (PCCT and S64) was 306 ± 244 days with a maximal span time of 910 days and a minimal time span of 9 days.

### 3.2. Qualitative Image Analysis

In terms of overall image quality, there was no significant difference between the EID-CT and PCCT images of the whole lung, though the PCCT (QIR+) images were rated best with a mean of 4.4 ± 0.7. The PCCT (QIR+) images of the whole lung were rated significantly better compared to the PCCT (QIR−) images (*p* = 0.004). Image sections were rated best in the PCCT (QIR+) images with a mean of 4.0 ± 0.8. In contrast to whole lung axial slices, enlarged lesion PCCT (QIR+) image sections were rated significantly better than the EID-CT and PCCT (QIR−) image sections (*p* < 0.001). Detailed results are given in Table 1 and displayed in Figure 2.

Image sharpness was rated best on the PCCT (QIR+) images with a mean of 4.2 ± 0.8 for whole lung axial slices and with a mean of 3.9 ± 0.8 for image sections. Both PCCT (QIR+) images were rated significantly better compared to the EID-CT images (image section: *p* < 0.001, whole lung: *p* = 0.014). Additionally, the PCCT (QIR+) images of enlarged lesions were rated significantly better compared to the PCCT (QIR−) images (*p* = 0.020) (Table 2 and Figure 2).

Image noise in image sections was rated best on the PCCT (QIR+) images with a mean of 3.5 ± 0.9, which proved to be significantly better compared to that of the EID-CT (*p* = 0.005) and PCCT (QIR−) images (*p* < 0.001). In contrast, image noise on whole lung axial slices was rated best in the EID-CT images with a mean of 4.1 ± 0.9. However, a significant difference was only seen in comparison to the PCCT (QIR−) images (*p* < 0.001). The exact results are summarized in Table 3 and displayed in Figure 2.

### 3.3. Quantitative Image Analysis

The SNR of the lung parenchyma was highest in the EID-CT images with a mean of 7.3 ± 2.4, which is significantly higher compared to that of the PCCT scans (*p* < 0.001). The SNR of the PCCT (QIR+) images with a mean of 5.2 ± 1.1 was significantly higher than the SNR of the PCCT (QIR−) images (*p* < 0.001). The lowest noise was found on the EID-CT images, with a mean of 131.7 ± 40.3, which proved to be significantly lower compared to PCCT (QIR−) (*p* < 0.001) and PCCT (QIR+) (*p* < 0.001). The PCCT (QIR+) noise was significantly lower in comparison to PCCT (QIR−) (*p* < 0.001) as shown in detail in Table 3.

### 3.4. Radiation Dose

The DLP (85.9 ± 36.1 mGy·cm vs. 112.6 ± 20.6 mGy·cm; *p* < 0.001), the CTDIvol (2.5 ± 1.1 mGy vs. 3.0 ± 0.6 mGy; *p* = 0.007) as well as the effective dose (1.4 ± 0.6 mGy vs. 1.9 ± 0.5 mGy; *p* < 0.001) of the PCCT scans were significantly lower compared to those of the EID-CT scans, as displayed in Table 4.

## 4. Discussion

This study examined the performance of the first clinical PCCT scanner in terms of radiation dose parameters as well as qualitative and quantitative image quality of an LD-HR chest CT scan protocol and compared the performance to that of a conventional EID-CT scanner. The PCCT achieved an improvement in qualitative image quality, while using significantly less radiation dose. However, the parameters indicating the quantitative image quality of the PCCT (e.g., SNR) were slightly higher compared to those of the EID-CT scanner. Moreover, PCCT images that made use of a quantum iterative reconstruction algorithm proved to be superior in qualitative image quality compared to PCCT images without an iterative reconstruction algorithm, in particular when judging enlarged image sections that require a higher spatial resolution.

Previous studies have investigated the image quality in lung imaging of a PCCT scanner. Bartlett et al. examined the visualization capabilities of a prototype PCCT compared to a conventional EID-CT for small pulmonary structures. They included twenty-two patients who underwent dose-matched PCCT and EID-CT scans with an HR-CT protocol. The reconstructed images were reviewed and evaluated by two thoracic radiologists. In this study, PCCT lung imaging offered a significant improvement in the resolution of small lung structures without compromising nodule evaluation [23]. Sartoretti et al. evaluated the qualitative and quantitative image quality of the first clinical PCCT using different levels of quantum iterative reconstructions (QIR-1 to QIR-4) for low-dose and ultra-high-resolution imaging of the chest. Additionally, the global noise index and global SNR index were measured. Two readers evaluated image quality, image noise and image sharpness. The authors found that subjective image sharpness and overall image quality was best at the QIR-3 level, while subjective noise reduction was best at the QIR-4 level. The global SNR index improved with higher reconstruction levels [18]. Jungblut et al. investigated the capabilities of an artificial intelligence-based detection system for pulmonary nodules with the first clinically approved PCCT compared to a dose-matched EID-CT system using an anthropomorphic chest phantom. Further, subjective image quality was evaluated by two readers and objective image noise was measured. They found a significantly better image quality in PCCT images compared to EID-CT images at a comparable-to-lower image noise [24].

These results of the present study are in line with previously published data. In particular, PCCT images that make use of QIR algorithms were rated significantly better compared to EID-CT images in overall image quality and image sharpness [18,23,24]. Although the majority of previously mentioned studies used the QIR level 3 for iterative reconstruction, we used QIRs at level 2.

In the present study, the subjective image noise was rated lowest on PCCT (QIR+) images for enlarged lesion images reaching statistical significance. At the same time, EID-CT images showed the lowest subjective image noise level for images of the whole lung, though the difference between PCCT (QIR+) images and EID-CT images was not statistically significant.

Interestingly, the calculated SNR was significantly higher on EID-CT images compared to PCCT images and therefore differed from the results of the qualitative evaluation. To our knowledge, there is no study comparing the SNR achieved with a PCCT in low-dose lung imaging and the SNR achieved with an EID-CT. However, the results of the present study differ from previously published data. Decker et al. report significantly higher SNR values in low-dose PCCT scans of the abdomen compared to EID-CT scans. In their study, the PCCT patient group was compared to a BMI-matched EID-CT cohort [25]. Grunz et al. examined the qualitative and quantitative image capabilities of a PCCT scanner in the microarchitecture of the bone and compared the results to images from a dual-source EID-CT. They discovered a substantially higher SNR in PCCT scans [26]. Since the SNR values in these studies were calculated from measurements of different tissue types in the abdomen and skeletal system, SNR values might differ in lung parenchyma. One reason might be the higher spatial resolution that comes with PCD technology leading to a higher SD in CT attenuation in an inhomogeneous tissue such as the lung. The study by Bartlett et al. performed noise ratio calculations in lung parenchyma with a prototype PCCT and compared the results to EID-CT noise ratios. They found that the noise ratio was lowest in EID-CT scans and significantly higher in PCCT scans, especially when using a kernel with a higher spatial resolution [23]. These results are in line with the findings of the present study.

Since several studies indicate a higher SNR in PCCT scans compared to EID-CT scans at the same level of X-ray exposure, the consecutive lower electronic noise and optimal photon energy weighting can be used to further reduce the radiation dose in PCCT scans [14,25,26]. The study by Sartoretti et al. reported an average CTDIvol of 1 ± 0.6 mGy [18]. In comparison, a large study conducted by Demb et al. analyzed the radiation dose of 12,529 patients undergoing lung cancer screening from various institutions in the USA. They described a mean CTDIvol of 2.4 ± 2 mGy [27]. In the present study, we achieved an average CTDIvol of 2.5 ± 1.1 mGy, which is comparable to the study of Demb at al. However, we used a high-resolution protocol to achieve better image quality for the evaluation of interstitial lung disease. In contrast, Sartoretti et al. used an ultra-low-dose protocol for PCCT scans of the lung, while lung cancer screening as described by Demb et al. is focused on reducing the radiation dose as much as possible. Therefore, our results differ from these studies in terms of dose parameters. Indeed, we achieved a significant reduction of the effective dose compared to EID-CT scans.

Our study has certain limitations. The body weight of each patient was recorded immediately before the CT examination with the PCCT and used to calculate the effective radiation dose of the PCCT as well as the EID-CT. Therefore, we assumed a constant body weight over the short mean time span between the EID-CT and PCCT scans, which was also indicated by constant body circumferences in both CT scans. The number of patients was limited, because all patients that qualified for the present study must have had an LD-HR CT scan with the EID-CT in the past. Furthermore, the EID-CT scanner that was available for comparison at our institute was an older scanner model and may not depict a comparison to an up-to-date EID-CT scanner. In particular, no iterative reconstruction was available for the EID-CT scanner. It would be desirable to compare the image capabilities of a PCCT to a state-of-the-art EID-CT to more accurately examine the influence of the different detector technologies on image quality. However, we believe that the presented comparison shows the realistic improvements achieved by PCCT compared to an EID-CT scanner that is still widely used in clinical practice.

In summary, LD-HR PCCT examinations of the chest promise better image quality, while allowing a significant dose reduction of up to 35.7% compared to EID-CT scans. Future studies may confirm these results with a comparison to an up-to-date EID-CT scanner and may investigate the diagnostic impact of improved image quality in PCCT.

## Figures and Tables

**Figure 1 diagnostics-12-01441-f001:**
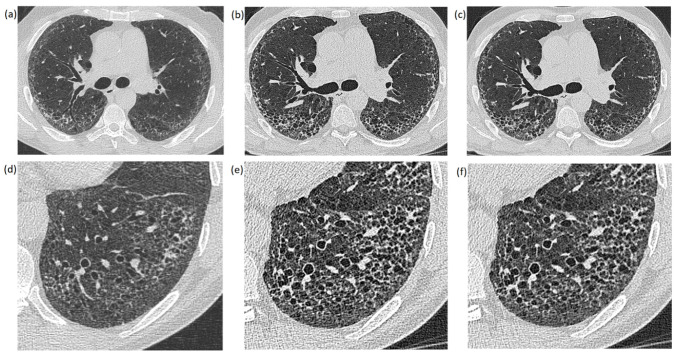
Images of the whole lung (**a**–**c**) and enlarged image sections (**d**–**f**) of the same patient: (**a**,**d**) were created with the EID-CT scanner; (**b**,**e**) were created with a PCCT scanner without an iterative reconstruction algorithm (QIR−); and (**c**,**f**) are PCCT images reconstructed with an iterative reconstruction algorithm (QIR+).

**Figure 2 diagnostics-12-01441-f002:**
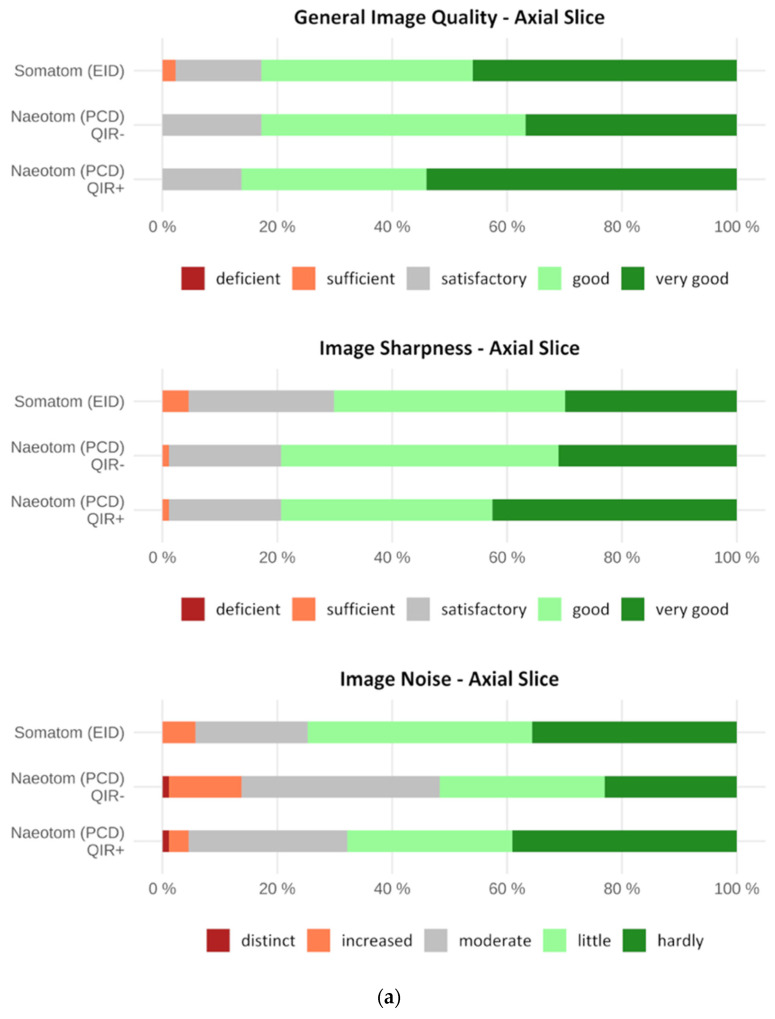
(**a**) Horizontal stacked bar chart showing the qualitative image evaluation of whole lung images. General image quality, image sharpness and image noise were assessed using a 5-point Likert scale. General image quality and image sharpness were rated best in PCD-CT (QIR+) images by the readers. In contrast, image noise was rated best in EID-CT images (without iterative reconstruction). (**b**) Horizontal stacked bar chart showing the qualitative image evaluation of image sections. General image quality, image sharpness and image noise were assessed using a 5-point Likert scale. General image quality, image sharpness and image noise were rated best in PCD-CT (QIR+) image sections by the readers.

**Table 1 diagnostics-12-01441-t001:** Mean values with standard deviation (SD) for overall image quality on EID, PCCT (QIR−) and PCCT (QIR+) images of enlarged lesions and the whole lung using a 5-point Likert scale as well as their associated *p*-values. Mean values are based on the rating on the Likert scale assessed by the readers (1—deficient, 2—sufficient, 3—satisfactory, 4—good, 5—very good). Statistical significance was assumed for *p*-values < 0.05.

	EID	PCCT (QIR−)	PCCT (QIR+)	Stat. Significance (*p*)		
	Mean SD	Mean SD	Mean SD	EID vs. PCCT (QIR−)	EID vs. PCCT (QIR+)	PCCT (QIR−) vs. PCCT (QIR+)
Lesion	3.4 ± 1.1	3.7 ± 0.8	4.0 ± 0.8	0.026	<0.001	<0.001
Whole Lung	4.3 ± 0.8	4.2 ± 0.7	4.4 ± 0.7	0.481	0.141	0.004

**Table 2 diagnostics-12-01441-t002:** Mean values with standard deviation (SD) for image sharpness on EID, PCCT (QIR−) and PCCT (QIR+) images of enlarged lesions and the whole lung using a 5-point Likert scale as well as their associated *p*-values. Mean values are based on the rating on the Likert scale assessed by the readers (1—deficient, 2—sufficient, 3—satisfactory, 4—good, 5—very good). Statistical significance was assumed for *p*-values < 0.05.

	EID	PCCT (QIR−)	PCCT (QIR+)	Stat. Significance (*p*)		
	Mean SD	Mean SD	Mean SD	EID vs. PCCT (QIR−)	EID vs. PCCT (QIR+)	PCCT (QIR−) vs. PCCT (QIR+)
Lesion	3.3 ± 1.1	3.6 ± 0.8	3.9 ± 0.8	0.001	<0.001	0.020
Whole Lung	3.9 ± 0.9	4.1 ± 0.7	4.2 ± 0.8	0.236	0.014	0.167

**Table 3 diagnostics-12-01441-t003:** Mean values with standard deviation (SD) for image noise on EID, PCCT (QIR−) and PCCT (QIR+) images of enlarged lesions and the whole lung using a 5-point Likert scale as well as their associated *p*-values. Additionally, signal-to-noise-ratio (SNR) and noise values are shown. Mean values are based on the rating on the Likert scale assessed by the readers (1—distinct, 2—increased, 3—moderate, 4—little, 5—hardly). Statistical significance was assumed for *p*-values < 0.05.

	EID	PCCT (QIR−)	PCCT (QIR+)	Stat. Significance (*p*)		
	Mean SD	Mean SD	Mean SD	EID vs. PCCT (QIR−)	EID vs. PCCT (QIR+)	PCCT (QIR−) vs. PCCT (QIR+)
Lesion	3.1 ± 1.3	2.9 ± 0.9	3.5 ± 0.9	0.030	0.005	<0.001
Whole Lung	4.1 ± 0.9	3.6 ± 1.0	4.0 ± 0.9	<0.001	0.725	<0.001
SNR	7.3 ± 2.4	4.0 ± 0.9	5.2 ± 1.1	<0.001	<0.001	<0.001
Noise	131.7 ± 40.3	238.7 ± 49.6	183.4 ± 36.9	<0.001	<0.001	<0.001

**Table 4 diagnostics-12-01441-t004:** Mean values with standard deviation (SD) regarding height, weight and different dose parameters as well as their associated *p*-values. Since height and weight were recorded during PCCT scans only, these parameters were adopted for the dose calculation of EID-CT scans as well.

	EID	PCCT	Stat. Significance (*p*)
	Mean SD	Mean SD	EID vs. PCCT
Height in cm		172.8 ± 10.0	
Weight in kg		77.4 ± 16.6	
DLP in mGy·cm	112.6 ± 20.6	85.9 ± 36.1	<0.001
CTDI_vol_ in mGy	3.0 ± 0.6	2.5 ± 1.1	0.007
Effective dose in mSv	1.9 ± 0.5	1.4 ± 0.6	<0.001

## Data Availability

The data are available from the corresponding authors upon reasonable request.

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
