# Peer review of "Low-Dose High-Resolution Photon-Counting CT of the Lung: Radiation Dose and Image Quality in the Clinical Routine"

_diagnostics, 2022, doi:10.3390/diagnostics12061441_

Round 1
Reviewer 1 Report
This study evaluates the image quality of computed tomography (CT) lung scans acquired with standard energy-integrated CT against a new photon counting CT system, with and without iterative reconstruction. Results demonstrate notable improvements in the image quality, as measured by three practicing radiologists, even with a dose raduction of almost 40%. Particularly, the combination of PCCT with the QIR algorithm and a reduced dose represent a significant improvement in diagnostic outcomes for lung CT scans.
The manuscript is well written, with a clear presentation of results, and a significant finding to relay to the CT research community.
This reviewer suggests a few things to address before publication:
-The figures are not referenced in the text. Can you add references to the figures, where appropriate. Figure 2 in particular could use some interpretation in the text.
-In Fig. 1, how was the window and leveling determined? Was this same standard applied to all images presented to the radiologists? Please comment in the manuscript.
-Can the authors comment on the expected impact of the different detector configurations (spatial resolutions) provided by the two CT scanners?
-How were the EID-CT scans reconstructed? Could some sort of IR technique have been applied there as well?
And a few minor details (some of which may be artifacts of my pdf viewer):
Line 43, comma between reticulation, etc.
Line 44, high resolution does not need dash
Line 97, superscript 2 in mm2
Line 145, a < symbol appears to be missing
Line 151, a ± symbol may be missing
Line 193, significantly
Line 249, significantly
Line 254, substantially
Line 267, a ± symbol may be missing
Line 270, a ± symbol may be missing
Line 293, 35.7% instead of 35,7% for consistency
Reviewer 2 Report
This manuscript focuses on radiation dose and image quality for low-dose high-resolution photon-counting CT of lungs. There are some issues.
1.In line 105, the quantum iterative reconstruction algorithm (QIR) is mentioned. Introduction to the QIR is necessary.
2.In line 123, the effective dose (ED) is mentioned. How to calculate the ED?
3.In lines 130-131, what are the criteria for the five image quality levels (1–deficient, 2–sufficient, 3–satisfactory, 4–good, 5–very good)?
4.In lines 131-132, what are the criteria for the five image noise levels (1–distinct, 2–increased, 3–moderate, 4–little, 5–hardly)?
5.In lines 132-133, what are the criteria for the five image sharpness levels (1–deficient, 2–sufficient, 3–satisfactory, 4–good, 5–very good)?
6.In lines 133-134, what is "the images were presented anonymized and randomized"?
7.In line 151, what is "305.5 243.6 days"?
8.In Table 1, what is the unit or meaning of the mean values with standard deviation?
9.In Table 1, what is the definition of "p"?
10.In line 271, "2.5 ± 1.1" needs the unit "mGy".
In conclusion, major revision is necessary.
Round 2
Reviewer 2 Report
This revised manuscript is ready to be published.